# BENCHDEPTH: ARE WE ON THE RIGHT WAY TO EVALUATE DEPTH FOUNDATION MODELS?

## ABSTRACT

Depth estimation is a fundamental task in computer vision with diverse applications. Recent advancements in deep learning have led to powerful depth foundation models (DFMs), yet their evaluation remains focused merely on geometry accuracy. Given the fact that downstream tasks increasingly rely on depth as guidance, we present **BenchDepth**, a new benchmark that evaluates DFMs through five carefully selected proxy tasks: depth completion, stereo matching, monocular feed-forward 3D scene reconstruction, SLAM, and vision-language spatial understanding. Our approach assesses DFMs based on their practical utility in real-world applications and provides complementary information to traditional benchmarks. We benchmark *eight* state-of-the-art DFMs and provide an in-depth analysis of key findings and observations. Interestingly, our results reveal discrepancies between rankings on traditional geometric benchmarks and those on downstream tasks, suggesting that existing evaluation protocols do not fully capture the practical effectiveness of DFMs. This underscores the importance of BenchDepth as a complementary benchmark, bridging the gap between geometry-centric metrics and application-driven evaluation.

## 1 INTRODUCTION

Depth estimation plays a crucial role in various computer vision applications, from 3D scene reconstruction, autonomous driving, to robotics Zhang et al. (2023); Li et al. (2023b); Zhu et al. (2024); Szymanowicz et al. (2024). In recent years, deep learning-based approaches have significantly advanced the field, leading to powerful foundation models capable of generating high-quality depth predictions across diverse input domains Eigen et al. (2014); Bhat et al. (2023); Ke et al. (2024); Yang et al. (2024b); Ranftl et al. (2022); Wang et al. (2024a; 2025). However, despite these advancements, evaluating and comparing depth estimation models remains an open challenge Ge et al. (2024). Existing evaluation protocols primarily emphasize geometry accuracy, which does not necessarily reflect the utility of depth in real-world applications.

Meanwhile, downstream tasks increasingly rely on depth as guidance, emphasizing the need for an evaluation framework that can reveal a model's potential across various applications Park et al. (2024); Szymanowicz et al. (2024); Zhu et al. (2024); Jiang et al. (2025); Cheng et al. (2025). Traditional benchmarks focus on constrained numerical accuracy, overlooking how different models generalize when deployed in application-driven tasks Ge et al. (2024). This disconnect often leads to discrepancies: models that excel in geometric benchmarks may not perform as well when integrated into end-to-end frameworks for practical applications.

To address this gap, we propose a new approach for benchmarking depth foundation models. Rather than relying solely on traditional depth evaluation metrics, we use downstream tasks as proxy tasks for model evaluation. By design, our benchmark shifts the focus from numerical alignment-based metrics to application-driven performance, thereby providing a complementary perspective on model capability. This direction is inspired by the success of large language models (LLMs), vision language models (VLMs), and self-supervised learning methods Achiam et al. (2023); Li et al. (2023a); He et al. (2020); Oquab et al. (2023); Siméoni et al. (2025), where the evaluation is often based on downstream tasks.

To this end, we propose **BenchDepth**, a benchmark consisting of five downstream proxy tasks: stereo matching Xu et al. (2023), depth completion Park et al. (2024), monocular feed-forward

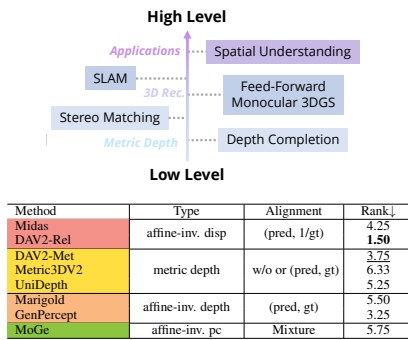

Figure 1: **BenchDepth illustration, results, and comparison with traditional benchmarks. Left**: Pearson correlation matrix between **BenchDepth** and the **standard benchmark** results. Proxy tasks exhibit strong internal consistency, indicating that BenchDepth captures meaningful shared structure across tasks and has the potential to generalize to other proxy tasks. Moreover, correlations between BenchDepth and traditional benchmarks are generally weaker or even negative, underscoring the gap between geometry-centric metrics and downstream utility. **Right**: We evaluate different types of depth predictions (highlighted with different colors) with proxy tasks in a bottom-to-top manner. We show the average rank of each depth method in the bottom right table. Different methods utilize different alignment strategies on the traditional benchmark, which is not necessary on BenchDepth.

3D scene reconstruction Szymanowicz et al. (2024), SLAM Zhu et al. (2024), and 3D-VQA Zuo et al. (2024). The tasks are selected in a bottom-to-top manner as shown in Fig. 1, ranging from applications in low-level to high-level vision. *Our goal is not to replace traditional metrics, but to complement them by revealing their limitations and proposing an application-driven benchmark that better reflects how DFMs are used in practice.*

In this paper, we benchmark *eight* state-of-the-art 3D foundation models with DepthBench. By examining their performance across proxy tasks, we provide new insights into what constitutes a good foundation depth model. *Notably*, our results reveal *discrepancies between rankings on traditional geometric benchmarks* Ge et al. (2024); Wang et al. (2024a) (We refer to the original papers for details about standard benchmark ranking) *and those on downstream tasks*, indicating that existing evaluation protocols do not fully capture the practical effectiveness of DFMs. Our main findings and conclusions are as follows:

1. Our correlation analysis (Fig. 1) shows stronger consistency among proxy tasks (*e.g.*, depth completion and SLAM: 0.88), indicating that the selected five tasks collectively form a representative and coherent benchmark. At the same time, correlations with traditional metrics are weaker or even negative, further emphasizing the gap between geometry-based evaluation and real-world utility.

2. Most depth foundation models improve the performance of downstream tasks, highlighting their potential for broader applications in the future.

3. Overall, DAV2 Yang et al. (2024b) achieves the best results across proxy tasks, demonstrating the benefits of scaling up training data and incorporating synthetic data.

4. Affine-invariant disparity methods consistently outperform other depth estimation approaches, even with MiDaS Ranftl et al. (2022) being the oldest method among them.

5. Despite being fine-tuned on a single dataset (Hypersim Roberts et al. (2021), synthetic), DAV2-Met significantly outperforms other metric depth models Hu et al. (2024); Piccinelli et al. (2024) trained on multiple datasets. This aligns with the conclusion of ZoeDepth Bhat et al. (2023) that fine-tuning a well-pretrained affine-invariant disparity model enhances metric depth estimation. Moreover, the performance gap suggests that incorporating synthetic data for metric depth training is crucial, as it allows models to learn high-frequency details that are often lost in real-world datasets Yang et al. (2024b); Li et al. (2024).

6. The performance improvement from Marigold Ke et al. (2024) to GenPercept Xu et al. (2024) underscores the importance of effective fine-tuning strategies for Stable Diffusion Rombach et al. (2022), a powerful foundation model. Expanding the training data could further unlock their potential, following the success of other methods, as the current fine-tuning process is limited to VKITTI Geiger et al. (2013) and Hypersim.

7. MoGe Wang et al. (2024a), as a novel approach for geometry estimation, demonstrates potential on DepthBench, though further research is needed to improve its performance.

8. For the highest-level task, VLM spatial understanding, all methods yield comparable results. This suggests that at this higher level, different depth estimation approaches can be equally effective.

We hope that our work will spark further discussion in the community about the best practices for depth model evaluation and pave the way for further research and development of depth estimation.

## 2 RELATED WORKS

### 2.1 DEPTH FOUNDATION MODEL (DFM)

Monocular depth estimation has seen significant advancements with the availability of large-scale public datasets Silberman et al. (2012); Geiger et al. (2012); Cordts et al. (2016), improved architectural designs Eigen et al. (2014); Li et al. (2023d); Bhat et al. (2021); Li et al. (2023c), and enhanced training strategies Chen et al. (2016); Fu et al. (2018); Li et al. (2022), *etc*. While earlier works primarily focused on achieving high performance in in-domain inference, the scaling of both models and datasets in deep learning Kaplan et al. (2020) has shifted recent research toward developing foundation models with strong zero-shot generalization across unseen domains (*i.e.*, diverse real-world images).

For example, MiDaS Ranftl et al. (2022) introduces a mixture-dataset training approach and adopts an affine-invariant disparity representation to handle cross-dataset inconsistencies. DAV2 Yang et al. (2024a;b) follows a similar formulation but scaled training further using a semi-supervised learning paradigm. Other works leverage the prior knowledge of Stable Diffusion Rombach et al. (2022) and fine-tune pretrained models for affine-invariant depth estimation Ke et al. (2024); Xu et al. (2024). Other lines of research such as Metric3DV2 Hu et al. (2024) and UniDepth Piccinelli et al. (2024) aim to predict metric depth by incorporating explicit camera models. MoGe Wang et al. (2024a) proposes a novel formulation using affine-invariant point maps Wang et al. (2024b) to represent monocular geometry. Despite the rapid progress in depth foundation models, a key challenge remains: how to evaluate and compare these models in a way that meaningfully reflects their effectiveness across diverse real-world applications.

### 2.2 EVALUATIONS OF DFMS

Eigen *et al* Eigen et al. (2014) introduced the first deep learning framework for monocular *metric* depth estimation, along with several standard evaluation metrics that remain widely used today. However, while depth estimation methods have diversified into various depth representations (as summarized in Tab. 1), existing works attempt to adopt the same evaluation protocol designed for metric depth estimation Ranftl et al. (2022); Yang et al. (2024b); Hu et al. (2024); Ke et al. (2024); Xu et al. (2024); Wang et al. (2024a).

In contrast, recent progress in other domains such as large language models (LLMs), vision-language models (VLMs), and self-supervised learning methods Achiam et al. (2023); Li et al. (2023a); He et al. (2020); Oquab et al. (2023); Siméoni et al. (2025)—demonstrates the importance of downstream-task evaluation for revealing the true potential of foundation models. Inspired by this, we propose an application-driven benchmark that assesses DFMs through five carefully selected proxy tasks, ranging from low-level to high-level vision.

Compared with previous benchmarks such as E3D-Bench Cong et al. (2025), which emphasizes multi-view geometry, and GeoBench Ge et al. (2024), which focuses on monocular depth estimation with traditional metrics, our benchmark shifts the emphasis toward downstream applications. By focusing on the monocular setting and evaluating depth estimation through real-world tasks,

BenchDepth provides a complementary perspective to geometry-centric evaluation and contributes to a more holistic understanding of depth foundation models.

## 3 BENCHDEPTH

We introduce **BenchDepth**, a novel benchmark for depth estimation based on carefully selected proxy tasks in a bottom-up manner (Fig. 1). Our design philosophy is to span a wide range of applications, from low-level tasks closely tied to depth prediction to high-level tasks where depth provides auxiliary guidance. This ensures that the evaluation reflects the practical utility of DFMs across diverse downstream scenarios.

### 3.1 TASK SELECTION

We group the proxy tasks into three levels:

**Low-level tasks: depth completion and stereo matching.** These tasks are closely related to metric depth estimation and differ mainly in their input prompts—sparse depth from sensors or stereo pairs with a fixed baseline. While the methods include task-specific components, we keep the architecture, training pipeline, and datasets strictly fixed across all experiments. The only variable is the input depth map from each DFM. As a result, performance differences serve as a fair and informative evaluation signal of the practical effectiveness of DFMs.

**Mid-level tasks: feed-forward 3D Gaussian Splatting (3DGS) and SLAM.** These tasks require more complex 3D reconstruction and differ in both representation (Gaussian splats vs. neural implicit) and input regime (single-view vs. multi-view), broadening the scope of our benchmark. Although less directly tied to DFMs than low-level tasks, recent studies have shown the growing use of DFM predictions as priors in these domains Szymanowicz et al. (2024); Zhu et al. (2024). By aligning architectures, training setups, and datasets, we ensure that observed performance differences can be attributed solely to the depth predictions of DFMs.

**High-level task: vision-language spatial understanding.** At the highest level, we evaluate the contribution of DFMs to VLMs Cai et al. (2024), where depth serves as a geometric prior for reasoning about 3D spatial relations. While performance differences are less pronounced here, the results reveal an important limitation: *current VLMs tend to rely on coarse layout cues, showing limited sensitivity to fine-grained depth errors.* Including this task highlights both the opportunities and the challenges of integrating depth into semantic reasoning systems, pointing to future research directions.

Together, these five tasks span different levels of abstraction, allowing users to focus on the evaluations most relevant to their applications.

### 3.2 MODEL SELECTION

Selected depth foundation estimation methods for benchmarking are summarized in Tab. 1. We choose the most representative methods from each depth estimation category. Note that though DAV2-Met Yang et al. (2024b), Metric3DV2 Hu et al. (2024), and UniDepth Piccinelli et al. (2024) are all metric methods, DAV2-Met is fine-tuned on a single metric dataset (Hypersim Roberts et al. (2021)), whereas the other two methods are trained with a mixture of many datasets. We use the default camera parameter assumption for Metric3DV2 and UniDepth. Since the original version of Marigold Ke et al. (2024) is hard to adapt to online training due to the large number of inference steps, we use the end-to-end fine-tuned version of Marigold Garcia et al. (2024) that supports one-step inference as a replacement.

For each proxy task, we use recent and well-integrated baselines selected for their compatibility with external depth inputs and representation diversity. DepthPrompting Park et al. (2024), Flash3D Szymanowicz et al. (2024), NICER-SLAM Zhu et al. (2024), and SpatialBot Cai et al. (2024) are all representative methods that explicitly incorporate DFMs in their design. IGEV Xu et al. (2023) does not use DFMs directly but serves as an important baseline for subsequent DFM-integrated stereo models such as FoundationStereo Wen et al. (2025) and DEFOM-Stereo Jiang et al. (2025).

Table 1: **Benchmark with metric depth completion.** We select DepthPrompting Park et al. (2024) as the baseline method and apply depth predictions from various foundation models as the guidance. We use different amounts of sparse samples (from 100 to 1) in this experiment. Best results are in **bold**, second best are underlined. *imp. (%)* indicates the average improvement ratio, and *rank* is calculated based on it. w/o depth refers to the baseline with only GT sparse depth as guidance.

| Method | 100 | | 32 | | 8 | | 4 | | 1 | | *imp.* | *rank* |
|---|---|---|---|---|---|---|---|---|---|---|---|---|
| | RMSE | MAE | RMSE | MAE | RMSE | MAE | RMSE | MAE | RMSE | MAE | | |
| w/o depth | 0.206 | 0.102 | 0.334 | 0.199 | 0.486 | 0.340 | 0.514 | 0.370 | 0.550 | 0.406 | - | - |
| Midas | 0.204 | 0.114 | 0.294 | 0.182 | 0.449 | 0.311 | 0.493 | 0.355 | 0.556 | 0.414 | +3.09 | 4 |
| DAV2-Rel | **0.191** | **0.099** | **0.279** | **0.166** | **0.427** | **0.292** | **0.471** | **0.336** | 0.533 | 0.396 | +9.26 | 1 |
| DAV2-Met | 0.202 | 0.112 | 0.287 | 0.178 | 0.431 | 0.297 | 0.472 | 0.338 | **0.529** | **0.392** | +6.48 | 2 |
| Metric3DV2 | 0.216 | 0.128 | 0.306 | 0.195 | 0.454 | 0.317 | 0.497 | 0.359 | 0.557 | 0.415 | -0.38 | 8 |
| UniDepth | 0.210 | 0.122 | 0.296 | 0.187 | 0.438 | 0.308 | 0.480 | 0.349 | 0.540 | 0.404 | +2.97 | 5 |
| Marigold | 0.210 | 0.121 | 0.296 | 0.187 | 0.448 | 0.314 | 0.491 | 0.356 | 0.555 | 0.414 | +1.76 | 6 |
| GenPercept | 0.199 | 0.110 | 0.284 | 0.174 | 0.436 | 0.301 | 0.479 | 0.342 | 0.542 | 0.402 | +6.16 | 3 |
| MoGe | 0.210 | 0.124 | 0.295 | 0.188 | 0.444 | 0.312 | 0.489 | 0.355 | 0.558 | 0.417 | +1.53 | 7 |

Most importantly, all selected baselines provide official training code, and our benchmark is directly developed on top of their implementations. More details are presented in Sec. 3.4.

When developing BenchDepth, we intentionally fix all aspects of the downstream pipeline. We either follow the default integration strategy used in the original papers or apply the most straightforward approach Zhang et al. (2023); Xu et al. (2023). Importantly, we restrict evaluation to the *predicted depth maps* from DFMs, excluding intermediate features. This avoids unfair advantages from model-specific backbones and ensures that the only factor varying across experiments is the DFM output. Although DFMs are not explicitly optimized for these tasks, the consistent performance differences observed validate BenchDepth as a fair measure of their relative effectiveness.

## 3.3 SCALABILITY

We acknowledge that downstream performance can also depend on the choice of the task architecture. In this work, we evaluate one representative model per task. While this design provides a controlled comparison and keeps the benchmark tractable, we view BenchDepth as a *framework* rather than a fixed set of results. Future work could include additional architectures, tasks such as video depth estimation or surface reconstruction, and stronger baselines (*e.g.*, 3DGS-based SLAM).

Our evaluation metrics are reported per task, reflecting the standards commonly used in each domain (*e.g.*, EPE for stereo, PSNR for view synthesis). To provide a broader perspective, we also compute improvement-over-baseline percentages and average rankings across tasks (Fig. 1).

In addition, we conduct a correlation study between BenchDepth results and traditional benchmarks. Specifically, we compute the Pearson correlation matrix between the average improvement on our proxy tasks and the standard benchmark results. The analysis (Fig. 1) reveals two key observations: (1) proxy tasks exhibit strong internal consistency, with depth completion and SLAM showing a particularly high correlation (0.88), indicating that our benchmark captures meaningful shared structure across tasks and has the potential to generalize to other proxy tasks; (2) correlations between proxy tasks and traditional benchmarks are generally weak and even negative, underscoring the gap between geometry-centric metrics and downstream utility. More details about this correlation study are presented in Sec. 4.

BenchDepth thus establishes a starting point for application-driven evaluation of DFMs. While not exhaustive, it demonstrates that DFMs have measurable and consistent impacts across diverse downstream settings, and highlights both their strengths and limitations when applied in practice.

## 3.4 DETAILS

Below, we present the five proxy tasks in detail and describe the modifications applied to selected methods to support depth evaluation using DepthBench. We use 8 GPUs to conduct the benchmark.

**Depth Completion:** Given sparse metric-scale depth measurements from sensors (*e.g.*LiDAR, Radar) and corresponding images, depth completion aims to generate dense metric depth predic-

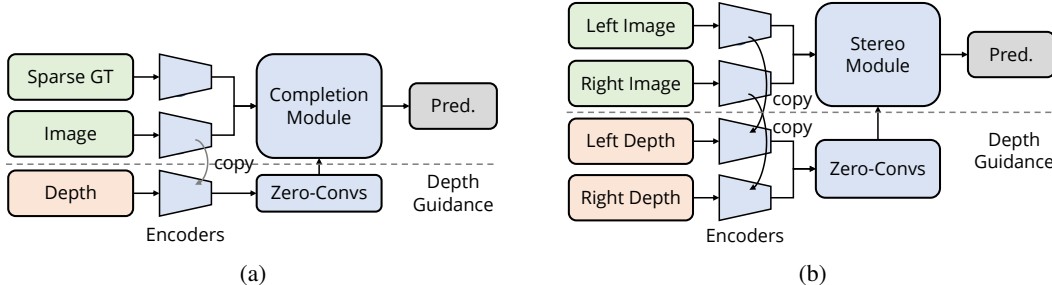

Figure 2: **(a)** Depth completion framework and **(b)** Stereo matching framework for depth benchmark. We adopt zero convolutions Zhang et al. (2023) to introduce depth guidance without modifying core components of proxy tasks.

tions. We select DepthPrompting Park et al. (2024) as the baseline method. While DepthPrompting enables the adaptation of foundation depth models for completion, its reliance on feature extractors from these models Li et al. (2023d) introduces bias, as the extractor quality may influence performance more than the predicted depth itself. To mitigate this, we standardize feature extractors across models and inject depth predictions using zero convolutions Zhang et al. (2023) (Fig. 2a). Additionally, we omit the alignment module in DepthPrompting to enable direct comparisons across depth methods. We use the NYU Depth V2 dataset Silberman et al. (2012) for this proxy task, following the official split with about 50k training samples and 654 testing samples.

**Stereo Matching:** This task estimates disparity from two images with a known baseline. Metric depth can be recovered from disparity using camera parameters. We adopt IGEV Xu et al. (2023) as our baseline and incorporate zero convolutions Zhang et al. (2023) to inject depth predictions as shown in Fig. 2b. Unlike prior works that develop task-specific strategies to integrate depth into stereo matching models Cheng et al. (2025); Jiang et al. (2025), our simple yet general approach allows for a more straightforward assessment of depth prediction quality. We use the SceneFlow dataset Mayer et al. (2016), which contains 35,454 training pairs and 4,370 test pairs with dense disparity maps. Middlebury 2014 Scharstein et al. (2014) and ETH3D Schops et al. (2017) are used for zero-shot evaluation.

**Feed-Forward Monocular 3DGS:** This task reconstructs scenes and synthesizes novel views from a single image using 3D Gaussian Splatting Kerbl et al. (2023). We use Flash3D Szymanowicz et al. (2024) as the baseline model. Flash3D incorporates a frozen depth foundation model in its first stage to estimate depth from the input image. The predicted depth and image are then processed by a UNet-like Ronneberger et al. (2015) network to estimate 3DGS parameters. Since the foundation depth model remains frozen and no features from the foundation model are used in the second stage, we can adopt different foundation models for the first stage and train Flash3D following the default recipe. We use the RealEstate10k dataset Zhou et al. (2018). It consists of real estate videos from YouTube, with 67,477 training scenes and 7,289 test scenes. Some outdated samples were removed, causing slight deviations from the results reported in Szymanowicz et al. (2024). The baseline result is obtained by directly using the officially released model with unmodified code.

**Simultaneous Localization and Mapping:** Simultaneous Localization and Mapping (SLAM) is a fundamental problem in computer vision with broad applications. We employ NICER-SLAM Zhu et al. (2024) as our baseline, as it integrates dense SLAM with a neural implicit representation for tracking and mapping from monocular RGB videos. Since NICER-SLAM can process RGB-D sequences, we replace the original sensor depth with depth predictions from different foundation models and train the system accordingly. To better assess the impact of depth predictions, we omit pseudo-depth loss during training. We evaluate models on the Replica dataset Straub et al. (2019), which provides RGB-(D) images rendered using the official renderer. All 8 scenes are used for benchmarking. For benchmarking, we replace the original input depth with estimated depth from different methods and omit the monocular depth loss (Eq. 13 in Zhu et al. (2024)), which depends on another depth model. We exclude Metric3DV2 since it was trained on this dataset, though there is no evidence of overfitting.

Table 2: **Benchmark with stereo matching.** We select IGEV Xu et al. (2023) as the baseline method and apply depth predictions from various foundation models as the guidance to fine-tune the baseline model.

| Method | SceneFlow | | Middlebury | | ETH3D | | *imp.* | *rank* |
|---|---|---|---|---|---|---|---|---|
| | EPE↓ | >3pt(%) ↓ | EPE↓ | >2pt(%) ↓ | EPE↓ | >1pt(%) ↓ | | |
| w/o depth | 0.496 | 2.599 | 0.857 | 6.655 | 0.283 | 3.575 | - | - |
| Midas | 0.483 | 2.502 | 1.061 | 7.316 | 0.273 | 3.383 | -3.07 | 7 |
| DAV2-Rel | **0.456** | **2.432** | **0.834** | 6.399 | 0.275 | **3.189** | **+5.77** | **1** |
| DAV2-Met | 0.471 | 2.473 | 0.938 | 6.177 | 0.270 | 3.698 | +1.46 | 5 |
| Metric3DV2 | 0.482 | 2.521 | 0.949 | 7.309 | 0.275 | 3.523 | -1.74 | 6 |
| UniDepth | 0.477 | 2.521 | 0.964 | 7.242 | 0.285 | 3.822 | -3.68 | 8 |
| Marigold | 0.475 | 2.499 | 0.899 | 6.519 | 0.273 | 3.485 | +1.87 | 4 |
| GenPercept | 0.473 | 2.485 | 0.935 | 6.649 | **0.265** | 3.374 | +1.99 | 3 |
| MoGe | 0.473 | 2.481 | 0.907 | **5.951** | 0.279 | 3.544 | +2.70 | 2 |

Table 3: **Benchmark with feed-forward monocular 3D scene reconstruction by novel view synthesis.** We select Flash3D Szymanowicz et al. (2024) as the baseline method and apply depth predictions from various foundation models as the model input. Following Szymanowicz et al. (2024), we present results of small, medium and large baseline ranges separately.

| Method | 5 frames | | | 10 frames | | | $\mathcal{U}[-30, 30]$ frames | | | *imp* | *rank* |
|---|---|---|---|---|---|---|---|---|---|---|---|
| | PSNR↑ | SSIM↑ | LPIP↓ | PSNR↑ | SSIM↑ | LPIP↓ | PSNR↑ | SSIM↑ | LPIP↓ | | |
| w/o depth | 24.285 | 0.803 | 0.151 | 21.767 | 0.729 | 0.203 | 21.241 | 0.705 | 0.230 | | |
| Midas | 24.964 | **0.812** | **0.125** | 22.290 | **0.735** | **0.179** | 21.769 | 0.710 | **0.212** | **+5.24** | **1** |
| DAV2-Rel | 24.965 | **0.812** | 0.129 | 22.305 | 0.733 | 0.185 | 21.703 | 0.706 | 0.218 | +4.21 | 3 |
| DAV2-Met | **25.000** | **0.812** | 0.128 | **22.341** | **0.735** | 0.182 | **21.842** | **0.711** | 0.215 | +4.81 | 2 |
| Metric3DV2 | 24.468 | 0.787 | 0.150 | 21.994 | 0.713 | 0.204 | 21.396 | 0.690 | 0.233 | -0.05 | 5 |
| UniDepth | 23.983 | 0.786 | 0.145 | 21.530 | 0.708 | 0.202 | 21.036 | 0.687 | 0.235 | -0.10 | 6 |
| Marigold | 23.974 | 0.779 | 0.162 | 21.515 | 0.701 | 0.219 | 20.952 | 0.676 | 0.248 | -4.19 | 8 |
| GenPercept | 24.119 | 0.787 | 0.140 | 21.489 | 0.705 | 0.197 | 21.029 | 0.682 | 0.230 | -0.14 | 4 |
| MoGe | 23.930 | 0.780 | 0.144 | 21.309 | 0.696 | 0.202 | 20.851 | 0.673 | 0.235 | -1.60 | 7 |

**VLM Spatial Understanding:** Vision-Language Models (VLMs) have demonstrated strong performance in 2D image understanding but remain limited in spatial reasoning Cai et al. (2024). Since depth maps contain spatial information, incorporating them as additional inputs may improve VLMs' 3D understanding. For this proxy task, we adopt SpatialBench Cai et al. (2024) to evaluate the impact of different depth models on VLM spatial reasoning. We use two VLMs: ChatGPT-4o and SpatialBot-Phi2-3B Cai et al. (2024). As ChatGPT-4o is not trained with depth, we render predicted depth maps with the magma colormap and provide paired textual prompts. In contrast, SpatialBot is jointly trained with paired images and depth maps. Although the released model is trained with ZoeDepth Bhat et al. (2023), it supports inference with estimated depth from any DFM, as confirmed in the Official Github Issue #12. We therefore use the official implementation to encode DFM-predicted depth maps and benchmark all models under the same setup.

## 4 BENCHMARK RESULTS

**Depth Completion.** Tab. 1 presents the benchmark results. DAV2-Rel Yang et al. (2024b) is the only method that consistently improves performance across almost all settings, achieving rank 1. Most methods provide a performance boost, except for Metric3DV2 Hu et al. (2024), which performs worse than the baseline. Interestingly, depth methods tend to be more beneficial when the available sparse ground-truth (GT) depth is limited. This suggests that foundation models provide useful guidance when GT depth is scarce. However, as GT depth increases, the ambiguity in selecting the appropriate depth source limits further improvements compared to using only sparse GT depth for guidance. In this case, some DFMs even lead to worse performance than the baseline, highlighting the strong dependence of this task on downstream-compatible DFMs.

**Stereo Matching.** Tab. 2 presents the results for stereo matching. In the in-domain setting, all foundation depth models significantly improve baseline performance, with an average 4.5% EPE gain. However, in zero-shot cross-domain evaluation, not all methods generalize well. DAV2-Rel, GenPercept Xu et al. (2024), and Marigold Ke et al. (2024) perform best. Metric depth models, such as Metric3DV2 Hu et al. (2024) and UniDepth Piccinelli et al. (2024), underperform compared to

Table 4: **Benchmark with Simultaneous Localization and Mapping (SLAM).** We select Nicer-SLAM Zhu et al. (2024) as the baseline method and apply depth predictions from various foundation models as the model input. acc and com are short for accuracy and completion, respectively. Rendered indicates that the input depth map is rendered by the dataset. We exclude Metric3DV2 and use gray for its results as it is trained with this dataset.

| Method | rm-0 acc↓ | rm-0 com↓ | rm-1 acc↓ | rm-1 com↓ | rm-2 acc↓ | rm-2 com↓ | off-0 acc↓ | off-0 com↓ | off-1 acc↓ | off-1 com↓ | off-2 acc↓ | off-2 com↓ | off-3 acc↓ | off-3 com↓ | off-4 acc↓ | off-4 com↓ | imp. | rank |
|---|---|---|---|---|---|---|---|---|---|---|---|---|---|---|---|---|---|---|
| w/o depth | 3.37 | 3.93 | 4.01 | 4.61 | 3.58 | 3.97 | 7.26 | 8.25 | 5.82 | 6.52 | 6.98 | 7.72 | 6.98 | 6.92 | 4.26 | 6.09 | - | - |
| Midas | 3.25 | 3.63 | 3.59 | 4.12 | 3.49 | 3.78 | 8.09 | 9.04 | 6.02 | 7.08 | 4.63 | **6.19** | **4.93** | 5.40 | 3.95 | **5.71** | +2.32 | 5 |
| DAV2-Rel | 3.30 | 3.92 | 3.52 | **3.85** | 3.28 | **3.59** | 6.16 | 6.94 | 5.78 | 6.62 | 6.55 | 7.09 | 7.00 | 6.43 | 4.26 | 6.09 | **+10.00** | **1** |
| DAV2-Met | 3.22 | **3.39** | **3.48** | 3.98 | 3.47 | 3.87 | 8.58 | 9.64 | 4.59 | 5.40 | 6.38 | 7.43 | 6.13 | 5.59 | 3.98 | 6.29 | +1.95 | 6 |
| Metric3DV2 | 3.48 | 3.64 | 3.45 | 3.93 | 3.73 | 4.09 | 9.55 | 10.53 | 5.82 | 6.41 | 5.20 | 6.67 | 6.73 | 6.78 | 4.51 | 6.65 | -4.19 | - |
| UniDepth | 3.11 | 3.49 | 3.73 | 4.38 | 3.80 | 4.06 | **5.96** | **6.91** | 5.05 | 6.05 | 6.48 | 7.41 | 5.83 | 5.95 | 4.60 | 6.76 | +7.08 | **2** |
| Marigold | **3.01** | 3.67 | 3.77 | 4.07 | 3.70 | 4.00 | 7.07 | 7.93 | 6.23 | 7.01 | 4.83 | 6.43 | 6.32 | 6.26 | 4.52 | 6.79 | +4.67 | 4 |
| GenPercept | 3.28 | 3.47 | 3.77 | 4.34 | 3.33 | 3.73 | 7.06 | 7.65 | **4.14** | **5.06** | **4.38** | 6.35 | 5.30 | **5.05** | 4.40 | 6.20 | +6.16 | 3 |
| MoGe | 3.26 | 3.67 | 3.67 | 4.23 | 3.89 | 4.33 | 8.86 | 9.83 | 4.55 | 5.58 | 5.68 | 6.73 | 6.40 | 6.32 | **3.92** | 5.98 | -4.04 | 7 |
| Rendered | 3.00 | 3.29 | 3.69 | 4.41 | 4.14 | 4.47 | 5.57 | 6.85 | 5.95 | 6.75 | 5.91 | 7.91 | 6.64 | 6.65 | 4.01 | 6.05 | - | - |

Text Prompt
We will provide you two images, the first one is the RGB image and the second one is the disparity image. For the disparity image, we use the magma colormap to render the disparity value. Deeper (farther) areas are depicted in black, transitioning through purple and pink, to the shallowest (closer) areas in bright yellow. The depth map can be inaccurate in some areas since it is predicted by a deep learning model. Please ignore this kind of mistake. Your task is to answer the following question by analyzing the image. Please use the depth map whenever necessary to provide more accurate and insightful answers.

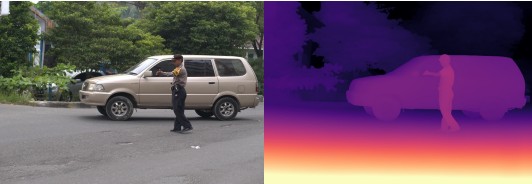

What is the spatial relationship between the police officer and the beige SUV?
(A) The police officer is in front of the beige SUV. (✓)
(B) The police officer is behind the beige SUV.
(C) The police officer is to the right of the beige SUV.
(D) The police officer is inside the beige SUV.
Answer: A ✓
The RGB image shows the police officer standing in front of the beige SUV. In the depth image, the police officer appears in darker colors compared to the SUV, indicating that the officer is closer to the viewer. Therefore, the officer is in front of the SUV.

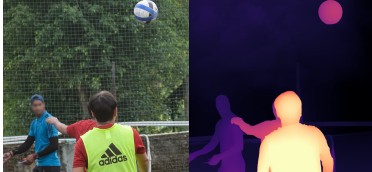

In the real world, what is the positional relationship of the man in red shirt and the man in blue shirt?
(A) The man in red shirt is to the left of the man in blue shirt.
(B) The man in red shirt is to the right of the man in blue shirt.
(C) The man in red shirt is in front of the man in blue shirt. (✓)
(D) The man in red shirt is behind the man in blue shirt.
Answer: D ✗
The disparity image shows the man in the red shirt in bright yellow, indicating he is closer to the camera. The man in the blue shirt is depicted in darker colors, suggesting he is farther away. Therefore, the man in the red shirt is behind the man in the blue shirt.

Figure 3: **Showcases of ChatGPT-4o on SpatialBot positional benchmark.** We highlight the text prompt describing rendered depth map in blue and mistakes made by ChatGPT-4o in red, respectively. In the first case, ChatGPT-4o correctly answers the question but misinterprets the depth map despite detailed prompts. As for the second one, despite correctly parsing the depth map, ChatGPT-4o provides an incorrect answer.

other types of depth estimation methods. Notably, DAV2-Met Yang et al. (2024b) outperforms other metric depth models, possibly benefiting from fine-tuning DAV2-Rel, despite being trained on only one dataset (Hypersim Roberts et al. (2021)). It somehow aligns with the conclusion in Bhat et al. (2023). The ability to predict sharper metric depth may also contribute to its superior performance.

**Feed-Forward Monocular 3DGS.** Tab. 3 shows the benchmark results. DAV2-Met achieves better performance compared with DAV2-Rel, suggesting that metric depth properties are beneficial for novel view synthesis tasks in real 3D environments. MiDaS Ranftl et al. (2022), despite being an older method, performs remarkably well with a rank of 1. DAV2-Rel also achieves strong results but slightly underperforms compared to MiDaS. Most metric depth methods, except for DAV2-Met and affine-invariant depth methods, fail to improve the baseline. Notably, this task exhibits the lowest correlation with the other proxy tasks, suggesting that it captures a complementary perspective on DFM quality that is not reflected by the other evaluation metrics.

**Simultaneous Localization and Mapping.** Tab. 4 presents the SLAM results. DAV2-Rel achieves the best results with a promising gap with other methods, indicating a superior potential for this task. UniDepth achieves the second-best results, highlighting the importance of metric depth for this task. GenPercept also obtains good results, possibly due to fine-tuning on Hypersim, a similar synthetic dataset. Nevertheless, the performance gap between GenPercept and Marigold still

Table 5: **Benchmark with spatial understanding of Vision Language Model (VLM).** We evaluate the effectiveness of depth predictions from various foundation models on the SpatialBench Cai et al. (2024). The *rank* column is omitted since all depth models perform similarly.

| Method | Pos.↑ | Exist↑ | Count↑ | Reach↑ | Size↑ | Method | Pos.↑ | Exist↑ | Count↑ | Reach↑ | Size↑ |
|---|---|---|---|---|---|---|---|---|---|---|---|
| ChatGPT-4o | 64.70 | 95.00 | 80.88 | 54.44 | 31.11 | SpatialBot | 61.76 | 75.00 | 92.41 | 51.67 | 28.33 |
| Midas | 62.74 | 90.00 | 80.26 | 54.44 | 37.22 | Midas | 55.88 | 55.00 | 92.41 | 46.67 | 30.00 |
| DAV2-Rel | 61.76 | 88.33 | 77.11 | 52.22 | 35.55 | DAV2-Rel | 55.88 | 60.00 | 93.13 | 46.67 | 30.00 |
| DAV2-Met | 61.76 | 86.66 | 80.44 | 59.44 | 38.88 | DAV2-Met | 55.88 | 65.00 | 93.13 | 45.00 | 28.33 |
| Metric3DV2 | 62.74 | 88.33 | 79.45 | 59.44 | 28.88 | Metric3DV2 | 58.82 | 55.00 | 93.13 | 50.00 | 28.33 |
| UniDepth | 64.70 | 93.33 | 80.55 | 62.22 | 37.77 | UniDepth | 58.82 | 60.00 | 92.41 | 53.33 | 28.33 |
| Marigold | 57.84 | 83.33 | 80.68 | 58.88 | 31.66 | Marigold | 55.88 | 60.00 | 93.13 | 46.67 | 30.00 |
| GenPercept | 60.78 | 85.00 | 81.03 | 57.77 | 37.77 | GenPercept | 55.88 | 65.00 | 93.13 | 48.33 | 28.33 |
| MoGe | 60.78 | 85.00 | 79.06 | 56.11 | 33.33 | MoGe | 55.88 | 60.00 | 93.13 | 50.00 | 28.33 |

highlights the effectiveness of its fine-tuning strategy. Interestingly, we also report results obtained from the ground-truth depth sensor provided by the dataset, and find that several DFMs even outperform this baseline. This promising result suggests that high-quality DFMs could serve as effective alternatives to traditional depth sensors in SLAM applications.

**VLM Spatial Understanding.** We use SpatialBench Cai et al. (2024) for this task. Unlike its original purpose of benchmarking different vision-language models (VLMs), we focus on evaluating the effectiveness of different depth estimations for the same VLM. We select ChatGPT-4o and Spatial-Bot Cai et al. (2024) as baseline VLMs, without and with depth inputs during training, respectively.

Surprisingly, for both VLMs, replacing ZoeDepth in SpatialBot with other DFMs does not significantly change performance, and ChatGPT-4o also shows little improvement when depth is added. All DFMs yield similar results, indicating no clear separation among models for this high-level spatial reasoning task. This saturation likely arises because VLMs are more sensitive to coarse layout cues and semantic structure, while being less responsive to fine-grained geometric detail. And most DFMs perform similarly well by providing sufficiently accurate coarse structures.

For this reason, we exclude VLM spatial understanding from the aggregated results, as it does not offer meaningful differentiation among DFMs. Nevertheless, we include qualitative results to highlight current limitations. Fig. 3 illustrates two examples from the positional benchmark in Spatial-Bench. In the first, ChatGPT-4o correctly answers the question but misinterprets the depth map despite detailed prompts, suggesting that training with depth signals is crucial for effective usage. In the second, ChatGPT-4o parses the depth map correctly but still produces an incorrect answer, underscoring the broader limitations of VLMs in reasoning about 3D space. These findings emphasize the importance of future research on how VLMs can better leverage depth beyond coarse structure.

**Correlation Analysis.** To further examine the representativeness of BenchDepth, we compute the Pearson correlation between improvements on our proxy tasks and metrics (delta and AbsRel) from traditional benchmarks. We use the benchmark from Lotus He et al. (2024) and the VGGT GitHub Issue #36. We include MiDaS, DAV2-Rel, Metric3DV2, Marigold, GenPercept, and MoGe in this calculation, as results for DAV2-Met and UniDepth are currently unavailable.

As shown in Fig. 1, proxy tasks exhibit strong internal consistency—for example, depth completion and SLAM show a high correlation of 0.88—suggesting that the selected tasks capture meaningful shared structure. In contrast, correlations between proxy tasks and traditional benchmarks are weak or even negative, indicating a clear gap between geometry-centric metrics and downstream utility. This finding further underscores the importance of BenchDepth as a complementary benchmark for evaluating DFMs.

## 5 CONCLUSION

We introduced **BenchDepth**, a benchmark for evaluating depth foundation models (DFMs) through downstream proxy tasks rather than alignment-based metrics. By benchmarking **eight** SoTA DFMs across depth completion, stereo matching, 3D scene reconstruction, SLAM, and vision-language spatial understanding, we provide a practical assessment of their effectiveness. Our experiments reveal key insights into the performance improvement of DFMs in real-world applications. We hope BenchDepth can assist the community in selecting DFMs for downstream applications.

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

# A    LARGE LANGUAGE MODELS USAGE

We used ChatGPT to polish the paper.

