# OpenReview forum: "BenchDepth: Are We on the Right Way to Evaluate Depth Foundation Models?"
_ICLR.cc/2026/Conference — ICLR 2026 Conference Withdrawn Submission_

### Official Review · Reviewer_aK22 · 2025-10-29

**Soundness:** 4
**Presentation:** 2
**Contribution:** 2
**Rating:** 4
**Confidence:** 4

**Summary:**

This paper introduces BenchDepth, a benchmark designed to evaluate depth foundation models (DFMs) through downstream tasks rather than traditional geometric metrics. The benchmark comprises five proxy tasks: depth completion, stereo matching, 3D scene reconstruction, SLAM, and vision-language spatial reasoning, aiming to capture the practical utility of DFMs in real-world applications. Eight recent DFMs are evaluated, and the authors report that model rankings differ significantly from conventional geometric benchmarks, highlighting a disconnect between numerical geometry metrics and downstream task effectiveness.

**Strengths:**

1. The paper targets an important and timely question regarding how to evaluate large-scale depth foundation models. Through experimental results, the paper identifies an evaluation gap between geometric accuracy and downstream performance on different tasks, which is an interesting observation.

2. The authors perform a large and comprehensive experimental study, covering multiple downstream tasks and eight modern DFMs. The empirical effort is substantial, and the collected results could serve as a useful reference dataset for the community. I appreciate the authors' effort in delivering the results.

**Weaknesses:**

1. The core research question is posed in the title (“Are we on the right way to evaluate DFMs?”). However, after carefully reading the paper, I find this question never brought up in the paper, never addressed in the paper, never answered in the paper, and never discussed in-depth in the paper. Sure, observations made by the paper (mismatch between current evaluation protocols and downstream performance) are valuable to this question, yet the discussion ends up right there without much high-level or technical insight. The paper reads as an experimental report rather than a proper research study. The paper states, "We hope that our work will spark further discussion (L119)", yet the discussion in the paper is minimal. No strong takeaways or principles explain why some DFMs perform better in certain downstream tasks.

2. Following 1, the technical depth is limited. The work mainly aggregates results from existing models and tasks without proposing new evaluation metrics, theories, or methodological innovations. Thus, the paper fails to clarify the intended contribution: is BenchDepth meant as a benchmark/tool or an analytical study? As a benchmark, details about usability, reproducibility, and release are missing. As an analytical study, it only contains shallow observations.

3. The writing and structure are weak. The introduction reads like related work, and the related work reads like an extension of the introduction. As 2. motivation and positioning are underdeveloped.

**Questions:**

1. What specific research question or hypothesis are you addressing?

2. Is BenchDepth publicly released and standardized for reproducible evaluation, or is it a one-time experimental setup?

3. How can future researchers extend or adopt this benchmark. Is there a unified scoring or evaluation pipeline?

4. Can you provide qualitative or conceptual insights explaining why DFMs behave differently across tasks?

---

### Official Review · Reviewer_fe6D · 2025-10-30

**Soundness:** 1
**Presentation:** 3
**Contribution:** 3
**Rating:** 4
**Confidence:** 4

**Summary:**

Paper proposes a novel benchmark for evaluating monocular reconstruction, based on five tasks. They perform correlation analysis over 8 models and 6 tasks, and show that four of them are strongly correlated and represent a coherent benchmark when averaged.

This is an interesting paper, and I like the direction, but it is incomplete.

The paper reports results for five results, one is excluded from the benchmark results in figure 1, and the two negative tasks are not reported.

The average of the positively correlated tasks is not reported, and presumably this is the actual benchmark. As such, no one can compare new methods on the benchmark.

**Strengths:**

Improved benchmarking in monocular reconstruction is very welcome.

The use of correlation to identify a coherent subset of tasks that make a benchmark is sensible.

Generally, I appreciate the design decisions, and I think this will make a valuable contribution to the literature when it is finished.

**Weaknesses:**

The coherence of the benchmark depends on the correlation analysis highlighted in Figure 1, and this is incomplete.

Task 5 in the text (VLM Spatial Understanding) is excluded from the figure, and the justification for excluding it doesn't really make sense. While the measure "Pos" shows little variation between the models, many of the other metrics show extremely large shifts.

Tasks 5,6,7 in Figure 1 (average, traditional delta, and traditional absrel) are not reported in the text.

In addition to this, the correlation analysis is performed using only 8 models. This doesn't automatically invalidate the statistics, but it means that the authors need to be very careful that one outlier isn't dominating the analysis. The authors need to generate plots for each pair of benchmarks in the analysis and confirm that the expected monotonic relationship exists.

One model is excluded from the SLAM benchmark table, Metric3DV2 and it is not clear how this was dealt with in the correlation analysis.

The negatively correlated benchmarks are also only reported on a subset of models, in other papers, and it's not clear how this altered the correlation analysis. The authors really need to bite the bullet and run all models on these benchmarks.

**Questions:**

The weakness contains a list of the issues I'm concerned with. It would be helpful if these were either addressed or you explained why they didn't matter.

On top of this, the writing needs tightening.

>Our correlation analysis (Fig. 1) shows stronger consistency among proxy tasks (e.g., depth
completion and SLAM: 0.88), indicating that the selected five tasks collectively form a
representative and coherent benchmark.

The correlation of selected measures, combined with the negative correlation of other measures, indicates that the benchmark is *not* representative of standard tasks in the literature. By no means is this a bad thing. Coherence is probably a better indication of benchmark usefulness. It's better to measure one thing well rather than a couple of things badly.

It's also not clear how meaningful the negative correlation in previous tasks is. Presumably, we could introduce trivial models such as constant depth, or even older models that perform badly on all tasks. This would then result in a positive correlation on all tasks.

Is this negative correlation just an example of a publication bias resulting in a collider bias? To be published, you need to be tuned to perform well on one of these types of task but not both, and that results in an apparent negative correlation.

---

### Official Review · Reviewer_AfUP · 2025-10-31

**Soundness:** 2
**Presentation:** 3
**Contribution:** 2
**Rating:** 2
**Confidence:** 4

**Summary:**

This paper introduces BenchDepth to evaluate depth foundation models based on downstream task performance, rather than traditional geometric accuracy metrics.
To assess how well DFMs perform in downstream tasks, the authors propose five proxy tasks: depth completion, stereo matching, feed-forward 3D reconstruction, SLAM, and vision-language spatial understanding.
Results show weak or even negative correlations between traditional geometric metrics and downstream task performance. The authors claim that geometry-centric evaluation may not reflect real-world effectiveness.

**Strengths:**

1. The paper shifts the benchmark focus from geometry-based metrics to application-driven performance, which brings a new insight for the community.
2. The benchmark covers both low-level and high-level depth tasks, which provides a multi-layered perspective on depth model utility.
3. The paper presents detailed quantitative analysis and interesting observations, such as strong internal consistency between proxy tasks but weak correlation with traditional benchmarks.
4. The multiple aspects of this benchmark could serve for evaluating DFMs beyond pure geometric fidelity for other works.

**Weaknesses:**

1. Depth estimation is fundamentally and inherently a metric prediction problem (Unlike LLMs and VLMs mentioned in line 49 – 52, whose evaluation rely on plausibility or relevance rather than quantitative accuracy). Thus, any evaluation suggesting the opposite conclusion must explain why geometric fidelity loses its predictive power. The authors fail to provide a compelling theoretical basis for this discrepancy, leaving open the possibility that the issue comes from their task integration rather than the metric itself.
2. Many downstream tasks such as SLAM and 3DGS contain complex neural components that can overshadow the role of depth inputs. As a result, the reported low correlation might reflect task noise rather than a true limitation of geometric benchmarks. Deeper analysis is still needed for this paper.
3. The benchmark only coversdoes not cover recent DFMs, e.g., CH3Depth [1]
[1] CH3Depth: Efficient and Flexible Depth Foundation Model with Flow Matching.
4. The paper is almost entirely empirical. The conceptual analysis of which geometric properties (such as scale, continuity, edge fidelity) actually matter for different tasks may build a bridge between geometric accuracy and downstream task performance, which is needed in this paper.
5. The authors themselves acknowledge that all DFMs perform similarly in VLM spatial understanding. This reduces the claimed significance of BenchDepth, as high-level reasoning tasks do not meaningfully discriminate model quality.

**Questions:**

1. Can you provide a principled explanation or analysis (beyond empirical observation) for the decoupling between geometric accuracy and downstream performance?
2. Have you tested, if replacing the depth maps with noisy or random ones yields similar weak correlations, to confirm that the benchmark truly measures depth utility rather than task variance? Many downstream networks may not be sensitive to the input depth maps. If that is the case, then even when provided with random or noisy depth maps, they might still produce similar outputs.

---

### Official Review · Reviewer_Xkaj · 2025-10-31

**Soundness:** 1
**Presentation:** 3
**Contribution:** 3
**Rating:** 2
**Confidence:** 4

**Summary:**

The authors presented a benchmark framework evaluating depth foundation models (DFMs) across a diverse set of downstream proxy tasks (depth completion, stereo matching, 3D Gaussian splatting reconstruction, learning-based SLAM, and VLM spatial reasoning). The benchmark aims to test practical utility rather than purely pixelwise accuracy, and the authors report that performance rankings under downstream tasks differ from those captured by standard depth metrics.

**Strengths:**

1. The work addresses a well-recognized gap: depth metrics (e.g., AbsRel, RMSE, δ thresholds) do not reliably predict whether a depth model is useful when integrated into 3D or geometric pipelines. Positioning evaluation around actual deployment tasks is well justified.

2. Diverse and well-chosen downstream tasks. The inclusion of tasks spanning low-level to high-level reasoning adds credibility to the claim that the benchmark captures general utility rather than performance tuned to a particular application.

**Weaknesses:**

1. Interpretation of the reported negative correlation between traditional metrics and downstream rankings requires clarification. The paper claims that standard depth metrics negatively correlate with downstream task performance. Taken at face value, this would imply that models performing worse under standard metrics could perform better in real applications, which is conceptually difficult to justify.

2. The benchmark only uses one baseline architecture per task. Performance may depend on model–pipeline interactions (e.g., certain DFMs may integrate more naturally into particular stereo or SLAM architectures). Using a single model per task risks overstating generality of the results.

3. There is a potential dataset overlap between DFMs’ training data and downstream task datasets. Some DFMs could have been trained or fine-tuned on datasets closely related to those used in the downstream evaluations. This may weaken conclusions, as models might benefit from dataset familiarity rather than intrinsic superiority as transferrable depth priors.

**Questions:**

1. Can you clarify on negative correlations between proposed benchmarks and standard metrics? This is my main concern that needs to be addressed.
2. Can you provide a table explicitly listing training datasets for each DFM and indicating where they overlap with each downstream task’s evaluation data?
3. Did you consider re-running one downstream task using an alternative baseline architecture to check whether performance rankings are stable across pipelines?

---

### Note · Authors · 2025-11-14

**Comment:**

We thank the reviewers for their comments.

**Withdrawal Confirmation:**

I have read and agree with the venue's withdrawal policy on behalf of myself and my co-authors.